# Research on 3D Path Optimization for an Inspection Micro-Robot in Oil-Immersed Transformers Based on a Hybrid Algorithm

**DOI:** 10.3390/s25092666

**Published:** 2025-04-23

**Authors:** Junji Feng, Xinghua Liu, Hongxin Ji, Chun He, Liqing Liu

**Affiliations:** 1State Grid Tianjin Electric Power Research Institute, Tianjin 300180, China; galago_@hotmail.com (J.F.); chun.he@tj.sgcc.com.cn (C.H.); liulq328@126.com (L.L.); 2College of Mechanical and Electronic Engineering, Shandong Agricultural University, Tai’an 271018, China; lxh9357@163.com; 3School of Electrical Engineering, China University of Mining and Technology, Xuzhou 221116, China

**Keywords:** path optimization, oil-immersed transformer, CTSP, A* algorithm, RRT, hybrid algorithm

## Abstract

To enhance the efficiency and accuracy of detecting insulation faults such as discharge carbon traces in large oil-immersed transformers, this study employs an inspection micro-robot to replace manual inspection for image acquisition and fault identification. While the micro-robot exhibits compactness and agility, its limited battery capacity necessitates the critical optimization of its 3D inspection path within the transformer. To address this challenge, we propose a hybrid algorithmic framework. First, the task of visiting inspection points is formulated as a Constrained Traveling Salesman Problem (CTSP) and solved using the Ant Colony Optimization (ACO) algorithm to generate an initial sequence of inspection nodes. Once the optimal node sequence is determined, detailed path planning between adjacent points is executed through a synergistic combination of the A algorithm*, Rapidly exploring Random Tree (RRT), and Particle Swarm Optimization (PSO). This integrated strategy ensures robust circumvention of complex 3D obstacles while maintaining path efficiency. Simulation results demonstrate that the hybrid algorithm achieves a 52.6% reduction in path length compared to the unoptimized A* algorithm, with the A*-ACO combination exhibiting exceptional stability. Additionally, post-processing via B-spline interpolation yields smooth trajectories, limiting path curvature and torsion to <0.033 and <0.026, respectively. These advancements not only enhance planning efficiency but also provide substantial practical value and robust theoretical support for advancing key technologies in micro-robot inspection systems for oil-immersed transformer maintenance.

## 1. Introduction

As critical components in high-voltage, large-capacity power systems, oil-immersed transformers constitute essential infrastructure in modern substations [1]. The transformer contains three primary internal elements, electromagnetic coils, laminated cores, and insulation (dielectric mineral oil, paper, etc.), which collectively enable efficient energy transfer through electromagnetic induction. Routine maintenance encompasses monitoring for structural anomalies including component displacement, core deformation, localized overheating, carbon traces from partial discharges, and dielectric degradation of insulation materials [2]. Conventional inspection methodologies requiring complete oil drainage present significant operational challenges, involving substantial resource expenditure and extended downtime. To address these limitations, micro-inspection robots have emerged as innovative solutions featuring compact dimensions, enhanced mobility, and precise controllability. These robots enable non-invasive internal assessments through onboard visual sensors while maintaining the transformer’s sealed environment. The implementation of such robots demonstrates substantial potential for optimizing maintenance efficiency, reducing operational risks, and improving grid reliability through predictive maintenance strategies.

The substantial spatial dimensions of oil-immersed transformers impose significant operational demands on inspection micro-robots, requiring them to traverse extensive spatial domains over prolonged durations. This necessitates the development of energy-efficient navigation algorithms capable of generating collision-free trajectories through the transformer’s labyrinthine interior.

Current research on robotics trajectory planning strategies often focuses on two-dimensional maps for collision-free shortest path planning from start to end points [3]. However, three-dimensional path planning is computationally complex and challenging to parameterize, making direct application of 2D strategies unfeasible in 3D spaces [4]. Traditional path-planning algorithms are categorized into sampling-based algorithms, such as Rapidly exploring Random Tree (RRT) [5] and Probabilistic Roadmaps (PRM) [6,7], which update paths from asymptotically optimal to globally optimal but suffer from inconsistencies due to randomness. Search-based algorithms, like A* [8] and Dijkstra’s algorithm [9], consistently find the optimal path between start and end points. Swarm intelligence and artificial intelligence algorithms, including genetic algorithms [10], Particle Swarm Optimization (PSO) [11], Ant Colony Optimization (ACO) [12], and Slime Mould Algorithm (SMA) [13], offer global search capabilities but may converge slowly or get trapped in local optima due to parameter sensitivity [14]. Typical algorithms of artificial intelligence are neural network algorithms [15], which have the characteristics of strong adaptive ability and better generalization, and are suitable for real-time decision-making application scenarios, but the large amount of data required and the high demand for computational resources mean these algorithms have a high threshold in applications [16]. In addition, the combination of different types of algorithms [17,18,19] to fully utilize the advantages of each algorithm is also a hot spot in academic research. For example, Zhang et al. [20] integrated the A* algorithm with the RRT algorithm, improving the timeliness and reliability of path planning for obstacle avoidance in driverless cars. Feng et al. [21] combined RRT algorithm and Artificial Potential Field (APF) algorithm to enhance the quality and stability of the optimal path generated by the RRT algorithm. Loc et al. [22] combined RRT algorithm with PSO, utilizing PSO algorithm to refine and enhance the initial paths planned by the RRT algorithm, thereby effectively avoiding falling into the local minimum state. Traditional FastSLAM algorithms suffer from particle degeneracy in environments with high similarity, such as internal oil-immersed transformers. To address this, Li et al. proposed a PSO-optimized FastSLAM framework that enhances localization accuracy by mitigating particle impoverishment [23]. Feng et al. proposed a monocular vision-based localization strategy utilizing multi-scale image enhancement and efficient pose estimation to fulfill real-time positioning requirements in harsh transformer oil environments [24].

The path-planning methods mentioned above primarily focus on optimizing the path between two points. However, the internal inspection micro-robot in the transformer needs to start from the starting point, route through a number of inspection points to capture images in sequence, and finally return to the starting point to complete the task. Therefore, these methods are not suitable for solving this type of problem. This issue is similar to the three-dimensional Traveling Salesman Problem (TSP) [25], which requires starting from a point, visiting multiple nodes and going back to the starting point while minimizing the total path length. Classical TSP solution methods mainly include the Held–Karp algorithm, greedy algorithm, genetic algorithm (GA), simulated annealing algorithm, and ACO [26,27]. Among these, ACO is widely used to solve TSP due to its high adaptability and robustness [28,29]. For example, Yan et al. [30] defined the multipoint planning problem with indoor path constraints as the Indoor Traveling Salesman Problem (ITSP) and solved it by combining Dijkstra’s algorithm and branch and bound (B&B) algorithm. Janoš et al. [31] solved the 2D indoor TSP problem using the SFF* algorithm. However, unlike the classical TSP, the transformer’s inspection points are subject to obstacles, making the inspection micro-robot’s path planning more complex than a classical TSP. When performing path planning for an inspection micro-robot in a transformer, the challenge is not only to optimize the total path length through multiple inspection points, but also to handle a complex three-dimensional environment with obstacles. This makes the problem more akin to a Constrained Traveling Salesman Problem (CTSP) and necessitates the adaptation of classical TSP solving methods to accommodate these environmental constraints. Classical TSP algorithms, such as the Held–Karp algorithm and the greedy algorithm, are usually very effective in solving the shortest path in the obstacle-free environments. However, in the presence of complex obstacles, these methods cannot directly address the issue of path feasibility. Genetic algorithms and simulated annealing algorithms, while possessing some global search capability, are prone to falling into local optimal solutions in high dimensional environments with numerous obstacles and may lack a consideration of path coherence. ACO, with its adaptivity and robustness, has become an effective tool for solving complex TSP problems [32].

Therefore, this paper proposes an innovative hybrid algorithmic framework aimed at solving the path-planning problem in a complex environment inside a large oil-immersed transformer. The task of multipoint traversal inside the transformer is transformed into CTSP and initial path planning is accomplished using ACO to determine the optimal sequence of inspection points. Building on this, a combination of path-planning techniques, including the A* algorithm, RRT and PSO, is constructed to accurately plan the paths between adjacent inspection points, ensuring that the inspection micro-robot’s movement avoids complex obstacles in the three-dimensional environment. This approach provides technical support for the inspection micro-robot to perform high-efficiency and low-consumption inspection tasks in the complex transformer environment.

## 2. Inspection Micro-Robots and 3D Environments in Transformers

### 2.1. Inspection Micro-Robots

Oil-immersed transformers account for 89.66% of China’s high-voltage power equipment (2023 industry statistics), demanding specialized inspection tools for internal defects. To address this issue, a 15 × 15 × 26 cm submersible micro-robot (as shown in Figure 1) with three-axis mobility has been developed for internal inspection in constrained transformer environments. It achieves floating/diving, hovering/positioning, and image acquisition functions. Featuring an elliptical titanium alloy shell designed for maneuverability in narrow winding gaps, the robot integrates ultrasonic positioning modules and propeller propulsion modules. When operating inside the transformer, the propeller propulsion modules provide thruster-based motion control, the ultrasonic range module achieves spatial localization, and the image acquisition module captures internal images. A dedicated control platform enables remote manipulation and collects defect images.

Compared to cubic designs (e.g., ABB’s TXplore), the elliptical geometry of the spherical submersible robot reduces hydrodynamic drag. It also enables agile navigation through <20 cm gaps between windings while maintaining stability under turbulent oil flows [33]. The micro-robot’s battery capacity supports continuous operation for 90 min, which has been validated to complete a full inspection cycle covering all 18 predefined inspection points within large oil-immersed transformers (180 MVA, 220 kV). The elliptical sealed structure and corrosion-resistant materials allow adaptable operation in various dielectric fluids such as mineral oil, synthetic esters, and silicone oils, which expands the scope of application scenarios for micro-robots.

**Figure 1 sensors-25-02666-f001:**
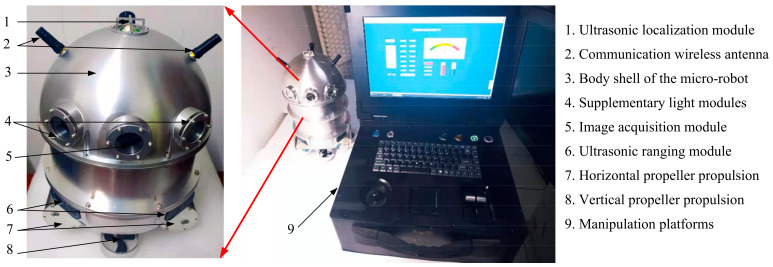
The inspection micro-robot for transformer internal inspection [34].

### 2.2. Three-Dimensional Environment Modelling in Transformers

The inspection micro-robot needs to work in the narrow and complex space inside the transformer and it is necessary to carry out reasonable planning for its inspection route. The 3D environment of the transformer is the basis of inspection micro-robot path planning. Therefore, it is very important to accurately construct the internal environment model of the transformer. Currently, there are several methods for environment modeling, such as topology method, visual graph method, free space method, and raster method, etc. This paper defines the internal transformer windings and fixed parts as impassable areas according to the free space method and the oil-filled area is defined as a free area in the 3D space. Path planning of inspection micro-robot is realized by Python (version 3.8.17) programming. The proposed system is validated for large oil-immersed transformers (180 MVA, 220 kV) with internal dimensions 6 × 3 × 3 m, representative of typical grid infrastructure.

In the transformer environment modeling, the O-XYZ coordinate system is constructed, with the dimensions of length, width, and height being (6000, 3000, 3000) mm. The three windings of A, B, and C phases are cylindrical, with radii of 800 mm and heights of 3000 mm, centered at (1100, 1500), (3000, 1500), and (4900, 1500), respectively. The internal 3D map of the transformer is shown in Figure 2.

The path-planning algorithm of inspection micro-robot can be described as follows. The 3D environment can be represented as C=[xc,yc,zc]T∈ℝ3. Known obstacles, the three-phase windings, can be described as Cobs=[xobs,yobs,zobs]T∈C. Cobs denotes the unreachable obstacle space and Cfree=C\Cobs denotes the free space. Based on the start point Pstart∈C, the goal pose Pend∈C, and environment C, the path algorithm generates the best path P within a limited duration. Path P consists of a series of feasible discrete ordered pose nodes set {Pi}i=1n∈Cfree, where P0=Pstart and Pn=Pend.

Let function $CollisionCheck(P,C_{obs})$ and BoundaryCheck(P,C) return the query result of whether the potential pose P∈C is legal, as shown in (1) and (2).(1)$$CollisionCheck(P,C_{obs})=\left\{\begin{array}{l} true,P_{i}\notin C_{obs}\\ false,P_{i}\in C_{obs} \end{array}\right.$$(2)$$BoundaryCheck(P,C)=\left\{\begin{array}{l} true,P_{i}\in C\\ false,P_{i}\notin C \end{array}\right.$$

## 3. Path-Planning Algorithms

To solve the challenge of guiding inspection micro-robots to perform inspection tasks efficiently and accurately in complex spaces, this paper explores several common and efficient path-planning algorithms, including the A* algorithm, RRT algorithm, and PSO algorithm. Each of these algorithms has distinct characteristics and can be used individually or in combination to form a more flexible and efficient path-planning strategy to optimize the inspection path inside the transformer.

### 3.1. A* Algorithm

The A* algorithm is an efficient heuristic search method commonly used in pathfinding and graph traversal problems. It combines the strengths of Dijkstra’s algorithm, which is adept at finding the shortest path, with the heuristic guidance of a greedy algorithm. By integrating the least-cost path search and heuristic evaluation, A* can quickly determine the optimal path between specified nodes.

The A* algorithm maintains two lists: the open list and the closed list. The open list contains nodes to be evaluated, while the closed list holds nodes that have already been processed. Initially, when the map is rasterized, the center of each square is considered a node and each node is classified as either walkable or unwalkable. The algorithm starts by setting point A as the initial node and adding it to the open list. The eight surrounding nodes of A are considered its child nodes and any walkable nodes among them are added to the open list for future examination. If a node is neither in the open list nor in the closed list, it indicates that the node has not yet been evaluated. Once a node has been fully evaluated and its path cost is determined, it is moved to the closed list. The algorithm iteratively processes nodes from the open list, selecting the node with the lowest cost and moving it to the closed list. This process continues until the target node is reached, at which point the path is saved, representing the shortest path found.

The cost function in the A* algorithm is defined as *f*(*n*):(3)f(n)=g(n)+h(n)(4)$$d_{ij}=\sqrt{{\left(x_{i}-x_{j}\right)}^{2}+{\left(y_{i}-y_{j}\right)}^{2}+{\left(z_{i}-z_{j}\right)}^{2}}$$
where *g*(*n*) represents the actual cost of moving from the start node to node *n* (i.e., the length of the path taken) and *h*(*n*) is the heuristic estimate of the cost to reach the goal from node *n* (typically calculated using either Euclidean or Manhattan distance). The total cost of moving from the start node to the goal node is represented by *f*(*n*). *d_ij_* represents the Euclidean distance between node *i* and *j*. The A* algorithm works by iteratively expanding the node with the lowest known cost and selecting the unexplored node with the smallest cost for further expansion, until the goal node is reached. This process ensures that the optimal path is found. The search process of the A* algorithm is illustrated in Figure 3 and the corresponding pseudocode is provided in Algorithm 1.

The A* algorithm is a heuristic graph search algorithm designed to find the shortest path between a start point and a goal on a predefined map. It is typically used for problems involving a single start and end point. However, in the case of transformer inspections, multiple waypoints must be visited, ultimately returning to the original start point—a scenario similar to the Constrained Traveling Salesman Problem (CTSP). The A* algorithm is not suited for solving such problems, as it cannot handle the complexities of planning a complete route that includes multiple points. Therefore, to optimize the inspection path that involves visiting several inspection nodes, it is essential to integrate other intelligent algorithms.
**Algorithm 1.** A* algorithm pseudocode.**Algorithm A***(start, goal)1: openSet ← {start}           //Initialize the open set with the start node2: fScore[start] ← HeuristicCostEstimate(start, goal)//Estimated total cost from start to goal3: **while** openSet is not empty do     //While there are nodes to process4:   current ← node in openSet with lowest fScore value//Select node with lowest estimated total cost5:   **if** current = goal then      //If goal is reached6:     return ReconstructPath(cameFrom, current)//Reconstruct and return the path7:   openSet ← openSet—{current}   //Remove current from open set8:   **for** each neighbor of current **do**  //Explore neighbors9:     tentative_gScore ← gScore[current] + Dist(current, neighbor)10:     **if** tentative_gScore < gScore[neighbor] then//Check if current path is better11:       cameFrom[neighbor] ← current//Record path12:       gScore[neighbor] ← tentative_gScore//Update cost13:       fScore[neighbor] ← gScore[neighbor] + HeuristicCostEstimate(neighbor, goal)//Update fScore14:       if neighbor not in openSet then//Discover new node15:         openSet ← openSet ∪ {neighbor}16:  **end for**17: **end while**18: **return** failure            //Return failure if no path is found

### 3.2. RRT Algorithm

The RRT algorithm is known for its ability to explore three-dimensional spaces. By using random sampling, it quickly generates multiple feasible paths, making it suitable for handling dynamic obstacles and complex spatial distributions. However, due to its random nature, there may be instances where it fails to find a path, and the smoothness and optimization of the path often require further adjustments.

The basic steps of the RRT algorithm are as follows:

1.Initialization: Create a tree T, with the root $q_{start}$ as the starting point.2.Sampling: Randomly generate a sample point $q_{rand}$, typically through uniform sampling in the space.(5)$$q_{rand}\sim \mathrm{Uniform}(X)$$
where *X* is the search space.3.Nearest Neighbor Search: Find the nearest node $q_{nearest}$ in the tree T to the sampled point $q_{rand}$.(6)$$d(q_{1},q_{2})=\sqrt{\sum_{i=1}^{n}{\left(q_{1i}-q_{2i}\right)}^{2}}$$4.Extension: Extend from $q_{nearest}$ towards $q_{rand}$ to generate a new node $q_{new}$. This step is usually constrained by a step size ε, i.e.,:(7)$$q_{new}=q_{nearest}+\varepsilon \frac{q_{rand}-q_{nearest}}{\left\|q_{rand}-q_{nearest}\right\|}$$5.Validity Check: Before adding the new node to the tree, check whether the path from $q_{nearest}$ to $q_{new}$ collides with any obstacles, i.e., whether it satisfies the collision test function in Equation (1). If there is no collision, add $q_{new}$ to the tree and record $q_{nearest}$ as its parent.6.Termination Condition: Repeat the above steps until a node sufficiently close to the goal region is found or the maximum number of iterations is reached.7.Path Generation: Trace back through the parent nodes from the goal node to construct a path from the start to the goal.

The schematic of the search process is illustrated in Figure 4 and the pseudocode for the RRT algorithm is shown in Algorithm 2.
**Algorithm 2.** RRT algorithm pseudocode.**Algorithm RRT**(x_start, x_goal, X_obs, Iter)1: V ← {x_start}         //Initialize the vertex set with the start node2: E ← ∅             //Initialize the edge set as empty3: **for** i = 1 to Iter **do**      //Iterate for a given number of iterations4:   x_rand ← Sample(i)     //Sample a random point in the space5:   x_nearest ← NearestNode(V, x_rand) //Find the nearest node to the random point6:   x_new ← Steer(x_rand, x_nearest)  //Move towards the random point from the nearest node7:   **if** NoCollision(x_nearest, x_new, X_obs) then//Check if the path to the new node is collision-free8:     V ← V ∪ {x_new}    //Add the new node to the vertex set9:     E ← E ∪ {(x_nearest, x_new)} //Add the new edge to the edge set10:   **end if**
11: **end for**
12: **Return** G = (V, E)       //Return the graph composed of the vertices and edges

### 3.3. PSO Algorithm

The PSO algorithm simulates swarm behavior for global search and can quickly approach the optimal path solution through the iterative updating of particles, demonstrating excellent performance in path optimization within complex environments. The pseudocode of the PSO algorithm is shown in Algorithm 3.
**Algorithm 3.** PSO algorithm pseudocode.**Algorithm PSO ( )**1: **Initialize** swarm of particles with random positions and velocities2: **for** each particle **do**3:   Calculate fitness of particle //Determine how good the current position is4:   Update personal best (pBest) if current fitness is better5: **end for**6: Determine global best (gBest) position among all particles//Find the best position found by any particle7: **while** stopping criteria not met **do** //Iterate until stopping condition is met8:   **for** each particle **do**9:      Update velocity based on pBest and gBest //Calculate new velocity10:     Update position based on velocity//Move particle to new position11:     Calculate fitness of particle      //Evaluate new position12:     Update personal best if current fitness is better 13:   **end for**14:   Update global best position if any particle is better//Update gBest15: **end while**16: **Return** gBest as the best found solution     //Return best solution found

### 3.4. Transformer Internal Inspection Node Sequencing Optimization

The overall path planning for transformer internal inspection nodes can be considered a variation of the classical TSP. In this context, path planning not only requires determining the order of node visits but also necessitates avoiding obstacles, making it a CTSP. The TSP is a classical optimization problem where a set of *n* inspection points $C=\{C_{1},C_{2},\ldots ,C_{n}\}$ is given and the distance between each pair of points $d_{ij}$ is known. The objective of the multi-point inspection planning problem is to find an optimal solution that minimizes the total path length while traversing all inspection points and returning to the starting point. This problem can be mathematically described by Equation (8):(8)$$\min L=\sum_{i=1}^{n-1}d_{\left(C_{i},C_{i+1}\right)}+d_{\left(C_{n},C_{1}\right)}$$(9)$$d_{\left(C_{i},C_{i+1}\right)}=\sqrt{{\left(x_{i}-x_{i+1}\right)}^{2}+{\left(y_{i}-y_{i+1}\right)}^{2}+{\left(z_{i}-z_{i+1}\right)}^{2}}$$
where $C_{1}$ is the selected starting inspection point, $C_{i}(i=1,2,\ldots ,n)$ are the inspection points to be visited, and $d_{\left(C_{i},C_{i+1}\right)}$ is the path length between inspection points *i* and *i* + 1.

The Ant Colony Optimization (ACO) algorithm, proposed by Dorigo M. in 1991, is an effective algorithm for solving TSP. The ACO algorithm is a bio-inspired optimization method. When ants search for food, they release pheromones on the paths they travel. When more ants follow the same path, the accumulation of pheromones increases, attracting more ants to choose that path. Therefore, ACO algorithm is a positive feedback algorithm. In the context of transformer internal multi-target inspection path planning, which can be viewed as a constrained TSP, the ant colony algorithm can be used to initially optimize the sequence of inspection points. Subsequently, algorithms such as A*, RRT, and PSO can be employed to complete the overall path planning.

Assume the number of internal repair points in the transformer is *n*, and the number of ants is *m*. The three-dimensional space of the transformer is shown in Figure 1. The path optimization process of the ACO algorithm is as follows:1.Pheromone Initialization(10)$$\tau_{ij}(t=0)=\tau_{0}$$
where $\tau_{0}$ is the initial pheromone concentration on all paths, which is a small constant.
2.Heuristic Function
(11)$$\eta_{ij}=\frac{1}{d_{ij}}$$
where $d_{ij}$ is the distance between inspection points *i* and *j*, and $\eta_{ij}$ is the heuristic function of inverse of $d_{ij}$.

3.Path Selection Probability

The probability $p_{ij}^{k}(t)$ that an ant selects the next inspection point *j* is determined by the pheromone and heuristic information:(12)$$p_{ij}^{k}(t)=\frac{{\left[\tau_{ij}(t)\right]}^{\alpha }{\left[\eta_{ij}\right]}^{\beta }}{\sum_{k\in allowed}{\left[\tau_{ik}(t)\right]}^{\alpha }{\left[\eta_{ik}\right]}^{\beta }}$$
where $\tau_{ij}$ is the pheromone concentration from node *i* to node *j*, α is the pheromone heuristic factor indicating the influence of pheromone on the ant’s decision, β is the heuristic function factor indicating the influence of path length on the ant’s decision, and “*allowed*” is the search space for the ant.
4.Pheromone Update
(13)$$\tau_{ij}(t+1)=(1-\rho )\tau_{ij}+\Delta \tau_{ij}(t)$$
(14)$$\Delta \tau_{ij}(t)=\sum_{k=1}^{m}\Delta \tau_{\mathrm{ij}}^{k}(t)$$
(15)$$\Delta \tau_{\mathrm{ij}}^{k}(t)=\left\{\begin{array}{l} \frac{Q}{L_{k}},\mathrm{if}\;\mathrm{ant}\;k\;\mathrm{passed}\;\mathrm{through}\;\mathrm{the}\;\mathrm{path}\;(i,\;j)\\ 0,\;\mathrm{otherwise} \end{array}\right.$$
where ρ∈(0,1] is the evaporation rate representing the pheromone decay, $\Delta \tau_{ij}(t)$ is the pheromone increment, $\Delta \tau_{\mathrm{ij}}^{k}(t)$ is the pheromone increment of single ant *k*, Q is pheromone constant, $L_{k}$ is the path length of ant *k*.

5.Loop Until Termination Condition is Met

Each ant starts from a random node and selects the next node based on the probability formula until a complete TSP path is constructed. After all iterations, the best path is selected, providing an optimized sequence for node visits. Since the ACO algorithm typically converges quickly for classical TSP problems where nodes are unobstructed, it is less effective at handling obstacles. In real transformers, windings create obstacles along the path, making it impossible to solely rely on the ACO algorithm for effective obstacle avoidance. Therefore, after optimizing the node visit sequence using the ACO algorithm, it is necessary to incorporate the A*, RRT, and PSO algorithms to further optimize the path between nodes for obstacle avoidance.

### 3.5. Transformer Inspection Path Planning Based on the Hybrid Algorithm

There are two basic modes for inspection of transformers: targeted inspection of specific points and routine inspections involving multiple inspection points. While the A*, RRT, and PSO algorithms excel in single-target path planning from start to end, they have limitations when it comes to multi-target path planning. Therefore, this paper combines the ACO algorithm, which is more adept at handling multi-target planning, to initialize the node sequence for 3D path planning. This forms an initial optimal network that starts from the origin, passes through multiple inspection nodes, and returns to the starting point, thus determining the sequence of inspections. Subsequently, the A*, RRT, and PSO algorithms are employed for local path planning between nodes, ultimately forming a complete inspection path.

The specific steps are as follows:

Whenever two nodes *i* and *j* are selected, a path-planning algorithm A*, RRT, or PSO is invoked to calculate the shortest feasible path between *i* and *j*, taking obstacles into account. Obstacle avoidance in these algorithms is computed as a cost increment, expressed as:(16)$$\Delta_{ij}=f(j)-d_{ij}$$

This cost increment is then integrated into the ACO algorithm pheromone update and path selection weights to optimize the path while avoiding obstacles.

The increment cost of ant *k* traveling from path *i* to *j* is included in the pheromone update as a path cost adjustment term:(17)$$\Delta \tau_{\mathrm{ij}}^{k}(t)=\left\{\begin{array}{l} \frac{Q}{L_{k}+\Delta_{ij}},\mathrm{if}\;\mathrm{ant}\;k\;\mathrm{passed}\;\mathrm{through}\;\mathrm{the}\;\mathrm{path}\;(i,\;j)\\ 0,\;\mathrm{otherwise} \end{array}\right.$$

During each path decision, the dynamic obstacle avoidance path cost is used as an adjustment factor. This adjusted factor influences the selection process, leading to a modification of the traditional ACO algorithm’s probability formula for ants to select the next node, as shown in Equation (18):(18)$$p_{ij}^{k}(t)=\frac{{\left[\tau_{ij}(t)\right]}^{\alpha }{\left[\eta_{ij}\right]}^{\beta }e^{-\lambda \Delta }}{\sum_{l\in allowed}{\left[\tau_{il}(t)\right]}^{\alpha }{\left[\eta_{il}\right]}^{\beta }e^{-\lambda \Delta l}}$$

The pheromone evaporation term is also adjusted accordingly:(19)$$\tau_{ij}(t+1)=(1-\rho )\tau_{ij}+\sum_{k=1}^{m}\Delta \tau_{ij}(t)$$

Thus, by combining the ACO algorithm for initial node sequence planning and path-planning algorithms (A*, RRT, and PSO) for local obstacle avoidance, each iteration begins with ants starting from the origin. The next node is selected based on probability and then using A*, RRT, or PSO algorithm to compute the shortest obstacle-free path. After completing a path, the pheromone concentration on that path is updated based on the actual path cost. This iterative process continues until the termination condition is met. Finally, the path with the highest pheromone concentration is selected as the optimal solution, with the pseudocode shown in Algorithm 4 and the flowchart combining the ACO algorithm and the A* algorithm is shown in Figure 5. The hybrid algorithm’s 3D obstacle avoidance path planning leverages the advantages of both types of algorithms, enhancing obstacle avoidance capabilities and computational efficiency.
**Algorithm 4.** Hybrid algorithm pseudocode.**Algorithm Hybrid Algorithm**1: **Initialize** pheromone matrix with ones2: **for** iteration = 1 to n_iterations **do**3:    all_paths, all_costs ← empty lists4:    **for** ant = 1 to n_ants **do**5:      available_nodes ← all nodes except start and end6:      path, cost ← [start_node], 07:      **while** available_nodes is not empty **do**8:        current_node ← last node in path9:      probabilities ← compute_transition_probabilities10:       next_node ← select from available_nodes using probabilities11:     segment_path ← **a_star/rrt/pso**
12:       **if** segment_path is **not None then**13:          update path and cost with segment_path and its cost14:      complete path to end using **a_star/rrt/pso**15:      store path and cost if valid16:    update best_path and best_length if a better path is found17: **return** best_path, best_length

## 4. Results and Discussion

The problem of internal transformer inspection can be broadly classified into two categories: specific fault-target detection and routine inspections involving multiple predefined inspection points. This section analyzes the path planning of an inspection micro-robot inside a transformer for both types of inspection problems.

### 4.1. Specific Fault-Target Detection

When internal faults such as partial discharge, overheating, insulation aging, or short circuits occur in a transformer, they often cause abnormal heat generation, gas production, or vibration changes in specific areas within the transformer. Inspection micro-robots as a novel inspection tool, offer flexibility and autonomy, enabling them to enter the narrow spaces inside the transformer for detailed inspections. By integrating sensors and advanced algorithms, the robot can collect and analyze data in real-time, thereby enabling accurate detection of fault locations.

Given the complex 3D environment and narrow spaces inside the transformer, selecting an appropriate path-planning algorithm is crucial. Different path-planning algorithms have distinct characteristics in terms of efficiency, stability, and adaptability. To comprehensively evaluate the performance of various algorithms in the complex internal environment of transformers, three commonly used 3D space planning algorithms are compared: the search-based A* algorithm, the sampling-based RRT algorithm, and the swarm intelligence-based PSO algorithm.

In a large oil-immersed transformer, an abnormal discharge fault was detected. Based on preliminary acoustic detection, the fault location was estimated to be at coordinates (4900, 2700, 1500) mm. The inspection micro-robot entered the transformer from the entry point at coordinates (100, 100, 2900) mm. The straight-line distance between the start and end points is 5635 mm. The following are the simulation results obtained using the three algorithms.

(1)A* algorithm path-planning simulation result

The A* algorithm is a classical graph-based path-planning algorithm known for its efficiency and path stability. By combining path cost and heuristic evaluation functions, the A* algorithm can reliably find a low-cost path. For the current task, the A* algorithm demonstrated high stability across multiple trials, consistently finding a successful path with a length that remained around 8800 mm. Figure 6 illustrates the optimized path inside the transformer from different perspectives, where “elev” represents the “Elevation Angle” and “azim” represents the “Azimuth Angle”.

(2)RRT algorithm path-planning simulation result

The RRT algorithm constructs paths using random sampling. In this project, the RRT algorithm demonstrated a significant speed advantage, capable of quickly generating a feasible path. The simulation results showed that the paths generated by RRT were shorter than those of A*, ranging between 7200 mm and 8348 mm, as in Figure 7. However, the randomness of RRT makes its output paths unstable, sometimes failing to find a path or deviating from the optimal solution.

(3)PSO algorithm path-planning simulation result

The PSO algorithm is a swarm intelligence optimization algorithm often used for approximating global optimal solutions. Although PSO is theoretically suitable for complex multi-objective optimization problems, in this study, the complexity of the transformer’s internal environment caused PSO to occasionally fail in finding a feasible path. This instability makes the PSO algorithm less suitable for environments with narrow spaces and dense obstacles. As shown in Figure 8, the path generated by the PSO algorithm clearly deviates from the optimal path, with a length of 11,648 mm. Compared to the A* and RRT algorithms, the inconsistent performance of PSO renders it less advantageous in the current task environment.

Based on the above analysis, the A* algorithm has a clear advantage in terms of path stability and reliability, making it suitable for tasks requiring deterministic paths. RRT can be a quick choice for time-sensitive tasks. Although the paths are less stable, most results have relatively shorter path lengths. Due to its unstable performance, the PSO algorithm does not offer significant advantages in the current task environment. Therefore, selecting the appropriate algorithm (or combination of algorithms) based on specific task requirements can significantly enhance the efficiency and accuracy of inspections.

### 4.2. Multi-Point Inspection Path Planning

Internal faults in oil-immersed transformers are primarily classified into thermal and electrical faults. While there are many specific types, the most common internal faults during operation involve winding and core faults, which account for approximately 60% of total transformer failures. Winding faults are mainly caused by internal short circuits or external impacts during transportation, where the winding coils are subjected to electromagnetic forces or other mechanical stresses. After multiple stress events, the insulation may be damaged, leading to winding deformation or loosening of insulation pads, which in turn reduces the insulation level. This can result in inter-layer or turn-to-turn insulation breakdown. Core faults are mainly caused by debris such as welding slag or loose nuts on the core surface and the long-term operation may cause the insulating paint on the surface of the silicon steel sheets to peel off. This can lead to internal core short circuits, or when there are defects in the transformer’s internal structure or core damage, multiple earthing points in the core can form a closed loop, potentially causing core burnout or melt earthing strips [35].

Based on the above analysis, areas in the transformer where winding and core faults are likely to occur are considered key inspection points. Additionally, historical fault points within the transformer have a relatively higher probability of recurrence. Therefore, 18 mandatory inspection points, including both vulnerable fault points and historical fault points, are designated as path constraints for the transformer’s inspection micro-robot, as in Table 1.

The multi-point inspection problem in transformers is similar to TSP, but with the key difference that obstacles exist between inspection points, preventing the direct application of traditional TSP solutions. Based on the theoretical analysis in Section 3.4, this paper first employs the ACO to optimize the order of inspection points. Further path planning is then conducted using the A*, RRT, and PSO algorithms. The following are the calculation results, which are summarized in Table 2.

(1)ACO algorithm for pre-planning of the 3D TSP

As shown in Figure 9, the preliminary path planned by the ACO algorithm does not effectively avoid obstacles. Therefore, further planning is required to ensure that the path between nodes is collision-free and avoids the windings.

(2)Integration of A* algorithm with ACO pre-planning

To illustrate the role of CTSP pre-planning, a comparison is made between paths optimized by A* with and without pre-planning using the ACO algorithm. The simulation results are shown in Figure 10 and Figure 11. It can be seen that the path length with CTSP pre-planning is 34,228 mm, while the path length without pre-planning is 72,306 mm. The pre-planned path is 52.6% shorter, significantly improving inspection efficiency.

(3)Integration of RRT algorithm with ACO pre-planning

As shown in Figure 12, the path planned by the RRT algorithm contains many invalid paths in various directions due to the randomness of its parameters. This simulation results in a longer total path length of 38,531 mm, which is 4303 mm longer than the A* algorithm with ACO. Additionally, the path smoothness is lower, with higher curvature and torsion.

(4)Integration of PSO algorithm with ACO pre-planning

As shown in Figure 13, the path planned by the PSO algorithm oscillates near the nodes. The path length is 64,163 mm, nearly double that of the A* algorithm. In this project, the PSO algorithm does not offer significant advantages.

### 4.3. Path-Planning Result Evaluation

In the internal inspection task of large oil-immersed transformers, the combination of the A* algorithm and ACO provides a relatively optimal overall path. However, since the A* algorithm is fundamentally a grid-based path search method, i.e., path planning on a discrete grid, the paths it generates are typically in the form of right angles or zigzags. This path characteristic can restrict the actual movement of inspection tools. For the inspection micro-robot, which is designed to maneuver flexibly and fluidly in narrow spaces, the right-angle paths generated by A* require the robot to make sharp turns continuously, which not only affects movement efficiency but also increases mechanical wear and tear, energy consumption, and ultimately prolongs the inspection time.

Therefore, further path smoothing is essential. By smoothing the path, it becomes more fluid and natural, reducing the movement resistance of the inspection micro-robot in narrow spaces, allowing it to follow a path that is closer to a continuous curve. This not only improves inspection efficiency and accuracy but also reduces frictional losses during turns, helping to maintain equipment stability and safety during inspections. The necessity of path smoothing lies in providing the most suitable trajectory for the inspection micro-robot, maximizing its advantages in autonomous movement within complex environments.

Curvature is one of the metrics used to measure the degree of curve bending and a higher value indicates more severe bending and more frequent sharp turns. Torsion is a parameter that measures the degree of twisting of a spatial curve and a higher value indicates more pronounced twisting in a three-dimensional space. A smaller curvature and torsion mean that the path is easier to execute, with fewer sharp turns or excessive twists, resulting in higher stability for the inspection micro-robot.

(1)Curvature

Curvature is a variable that measures the degree of bending of a curve at a given point. The higher the curvature, the more the curve bends, and the more frequent the sharp turns. The formula for curvature is as follows:(20)$$\kappa =\frac{\left\|r^{\prime }(t)\times r^{\prime \prime }(t)\right\|}{{\left\|r^{\prime }(t)\right\|}^{3}}$$
where $r^{\prime }(t)$ is the first derivative of the curve (velocity vector) and $r^{\prime \prime }(t)$ is the second derivative.

(2)Torsion

Torsion describes the degree of twisting of a spatial curve. The higher the torsion, the more complex the twisting of the curve in space. The formula for torsion is:(21)$$\tau =\frac{\left(r^{\prime }(t)\times r^{\prime \prime }(t))\cdot r^{\prime \prime \prime }(t\right)}{{\left\|r^{\prime }(t)\times r^{\prime \prime }(t)\right\|}^{2}}$$
where $r^{\prime \prime \prime }(t)$ represents the third derivatives of the curve.

(3)Curve-Smoothing Method

B-spline curve is formed by piecing together polynomial segments and it is advantageous for smoothing paths because it can create a continuous and smooth curve [36]. By adjusting the control points, the shape of the curve can be modified and it offers local control, meaning that adjusting one control point only affects the corresponding local section of the curve. In this paper, the B-spline method is used to smooth the path optimized by the A* + ACO hybrid algorithm. The simulation results are shown in Figure 14. After B-spline interpolation and smoothing, the total path length is 37,766 mm. The curvature and torsion are shown in Figure 15. The curvature is less than 0.033, indicating that the curve is fairly smooth overall. The torsion falls within the range of −0.034 to 0.026, suggesting that the path does not exhibit excessive twisting in the three-dimensional space, meeting the mechanical requirements for the inspection micro-robot to navigate within the transformer.

(4)Path-Tracking Stability via Lyapunov Theory

To ensure the smoothed path meets closed-loop control requirements, formal stability guarantees are established through the following analysis:

Defining tracking errors between the inspection robot’s actual pose (*x*,*y*,*z*,*θ*) and the reference path (*x_d_*, *y_d_*, *z_d_*, *θ_d_*), the lateral error is *e_xy_*, height error *e_z_*, and the heading error is *θ_e_*.(22)$$e_{xy}=\sqrt{{\left(x-x_{d}\right)}^{2}+{\left(y-y_{d}\right)}^{2}}$$(23)$$e_{z}=z-z_{d}$$(24)$$\theta_{e}=\theta -\theta_{d}$$

For a differential-drive robot, the error dynamics derived from nonholonomic constraints are:(25)$$\left\{\begin{array}{l} {\dot{e}}_{xy}=v_{xy}\sin\theta_{e}\\ {\dot{e}}_{z}=v_{z}\\ {\dot{\theta }}_{e}=\omega -v_{xy}\kappa \end{array}\right.$$
where $v_{xy}$ and $v_{z}$ are linear velocity in the horizontal and vertical direction, respectively, ω is the angular velocity, and κ is the path curvature.

Lyapunov function candidate:(26)$$V(e,\theta_{e})=\frac{1}{2}e_{xy}^{2}+\frac{1}{2}e_{z}^{2}+\frac{1}{2}\theta_{e}^{2}$$

The objective is to design control law ω to drive $e_{xy}\to 0,e_{z}\to 0,\theta_{e}\to 0$:(27)$$\begin{array}{c} \omega =v_{xy}\kappa -k\theta_{e}\\ v_{z}=-k_{z}e_{z} \end{array}$$
where *k* > 0 and *k_z_* > 0 are the feedback gains of horizontal direction and the vertical direction, respectively. Combined with the error dynamics in Equation (25), the time derivative of *V* is:(28)$$\begin{array}{c} \dot{V}=e_{xy}{\dot{e}}_{xy}+e_{z}{\dot{e}}_{z}+\theta_{e}{\dot{\theta }}_{e}=e_{xy}(v_{xy}\sin\theta_{e})+e_{z}(v_{z})+\theta_{e}(\omega -v_{xy}\kappa )\\ =e_{xy}v_{xy}\sin\theta_{e}-k_{z}e_{z}^{2}-k\theta_{e}^{2} \end{array}$$

For small heading errors (*θ_e_* ≈ 0), Equation (28) can be simplified as Equation (29) using $\sin\theta_{e}\approx \theta_{e}$:(29)$$\dot{V}\approx v_{xy}e_{xy}\theta_{e}-k_{z}e_{z}^{2}-k\theta_{e}^{2}$$

Selecting parameters (*k*, *k_z_*) appropriately ensures and proves the convergence of all tracking errors (*e_xy_*, *e_z_*, *θ_e_*).

The A* + ACO raw path exhibits significant jaggedness, forcing the robot to perform abrupt turns at inflection points. This places excessive demands on the controller to respond promptly under speed/acceleration constraints, resulting in overshoot and large tracking errors. In contrast, the smoothed path with continuous curvature reduces dynamic stress on the controller, thereby minimizing tracking deviations. By applying the identical Lyapunov controller to both paths and simulating the robot’s motion trajectories with an initial velocity of 0.3 m/s and controller parameters *k* = *k*_z_ = 1.0, the simulation results in Table 3 demonstrate that the smoothed path achieves significantly lower tracking errors. This indicates that the actual control system operates closer to the theoretical assumptions (e.g., no actuator saturation, validity of linearization) during path following, thereby validating enhanced stability of the smoothed trajectory.

### 4.4. Discussion

Based on the analysis and simulation results presented, the proposed path planning method, which integrates the A* algorithm with ACO, has demonstrated superior performance in navigating an inspection micro-robot within large oil-immersed transformers. However, there are still areas where the performance can be further improved: (1) in practical applications, the internal environment of a transformer may undergo dynamic changes, such as fluctuations in oil flow or variations in equipment operating conditions, which can impact the robot’s navigation path. While the current algorithm performs well in pre-planning and path optimization, there is room for improvement in terms of real-time responsiveness to such environmental changes. Future research could focus on incorporating more dynamic path-adjustment mechanisms that allow the robot to quickly replan its path in response to sudden changes, without requiring complete recalculation, thereby enhancing the system’s responsiveness and robustness. (2) The internal environment of a transformer is not only complex and narrow, but also potentially hazardous. In addition to minimizing path length and ensuring smoothness, the robot must also consider path safety. For example, the robot needs to avoid areas with strong electromagnetic fields or high temperatures. Future studies could integrate environmental sensing and multi-modal information fusion technologies, incorporating physical field information (such as temperature and electromagnetic fields) into path planning. This would enhance the safety and stability of the planned paths, ensuring safer navigation for the robot.

## 5. Conclusions

This paper addresses the critical challenge of path planning for an inspection micro-robot operating within large oil-immersed transformers. An integrated approach that combines the strengths of multiple algorithms to achieve superior performance was constructed and optimized. This paper presents a novel and effective path-planning framework that significantly enhances the inspection efficiency and flexibility of miniature inspection robots.

1.Constrained TSP formulation and node pre-planning: To tackle the complex three-dimensional environment inside transformers, the problem is formulated as a constrained TSP. The ACO algorithm is employed for pre-planning the optimal visitation sequence of inspection nodes. This approach effectively reduces the overall path complexity and serves as a foundation for subsequent path planning.2.Integration of path-planning algorithms: Building upon the pre-planning stage, path-planning algorithms such as A*, RRT, and PSO are utilized to generate detailed paths. The A* algorithm, when integrated with the pre-planned node sequence, demonstrates the best performance in terms of path length, smoothness, and stability. The B-spline smoothing method is further applied to enhance the fluidity and naturalness of the robot’s movement path, addressing the limitations of traditional A* algorithms that tend to produce right-angle paths.3.Performance evaluation: The combination of A* and ACO achieves the shortest path length while maintaining excellent smoothness and stability. After B-spline smoothing, the total path length is 37,766 mm, with a curvature less than 0.033 and torsion within the range of −0.034 to 0.026, ensuring that the path is both smooth and free from excessive twisting. This meets the mechanical requirements for the miniature inspection robot to navigate within the transformer efficiently.

## Figures and Tables

**Figure 2 sensors-25-02666-f002:**
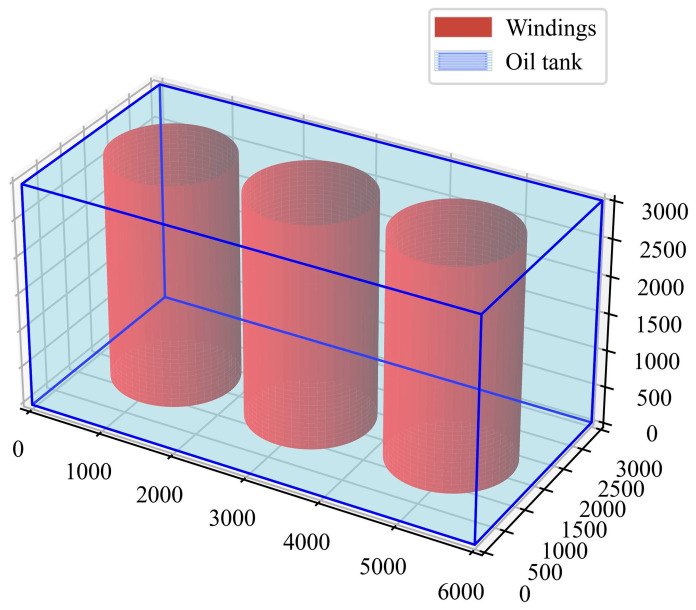
A 3D environment model of an oil-immersed transformer.

**Figure 3 sensors-25-02666-f003:**
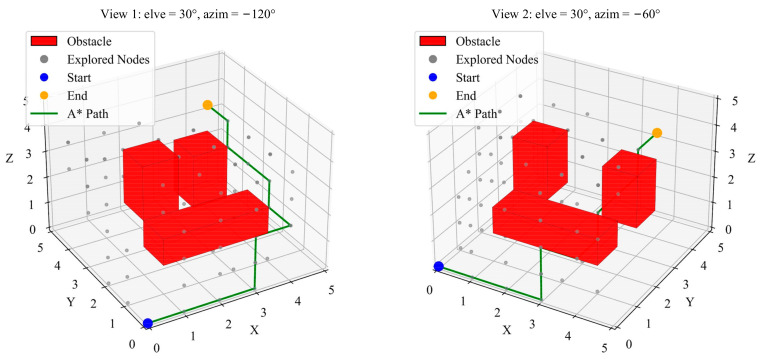
A* algorithm of the searching process.

**Figure 4 sensors-25-02666-f004:**
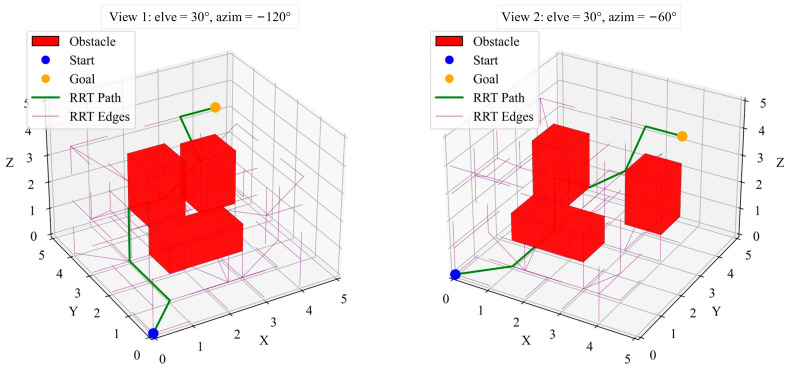
RRT algorithm of the searching process.

**Figure 5 sensors-25-02666-f005:**
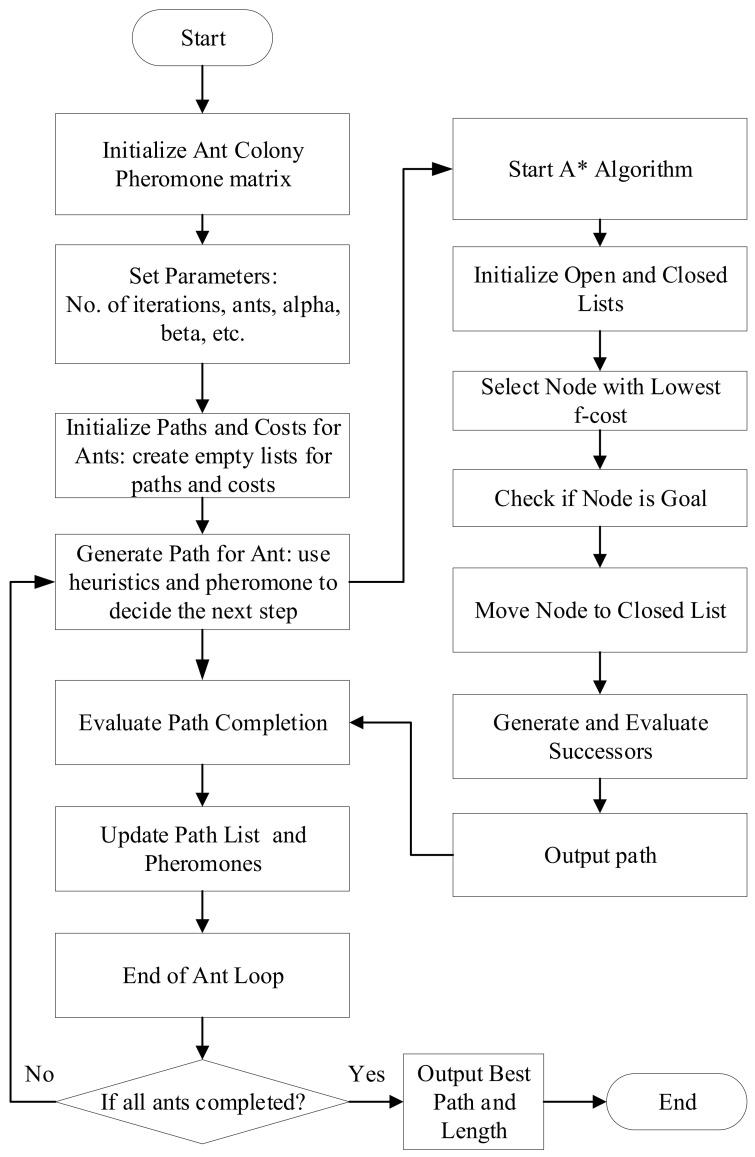
Hybrid algorithm based on the ACO and A* algorithm flowchart.

**Figure 6 sensors-25-02666-f006:**
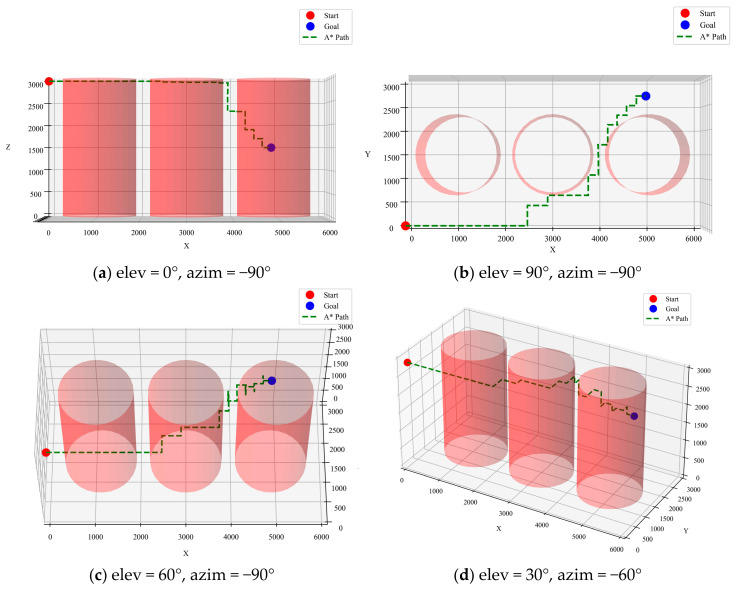
A* algorithm optimized path.

**Figure 7 sensors-25-02666-f007:**
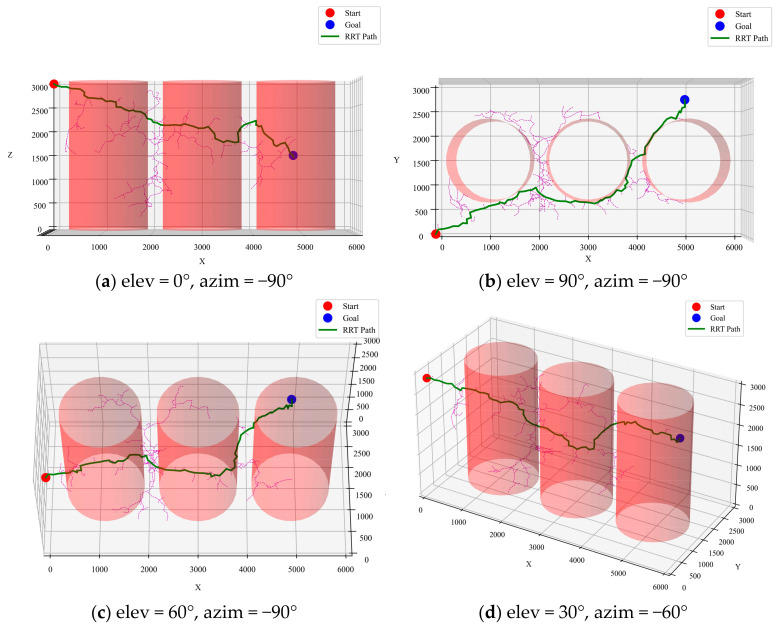
RRT algorithm optimized path.

**Figure 8 sensors-25-02666-f008:**
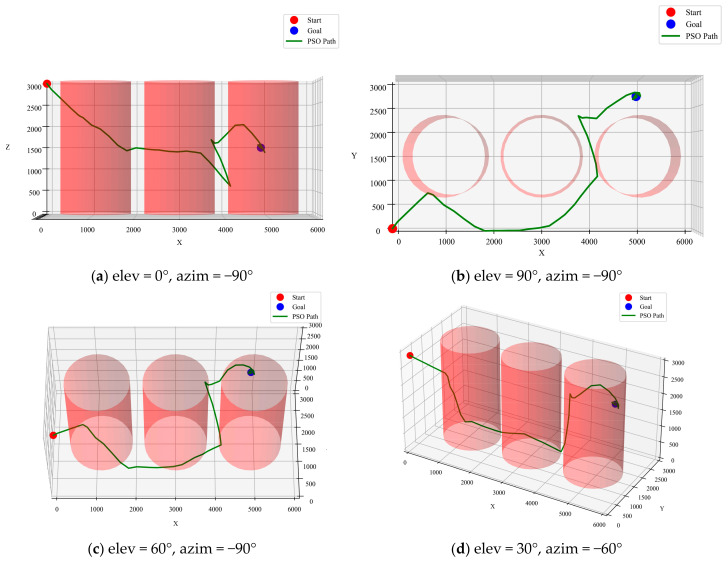
PSO algorithm optimized path.

**Figure 9 sensors-25-02666-f009:**
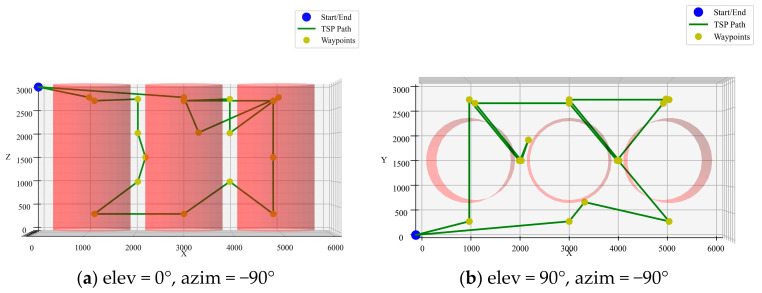
ACO preliminary planning of the path node order.

**Figure 10 sensors-25-02666-f010:**
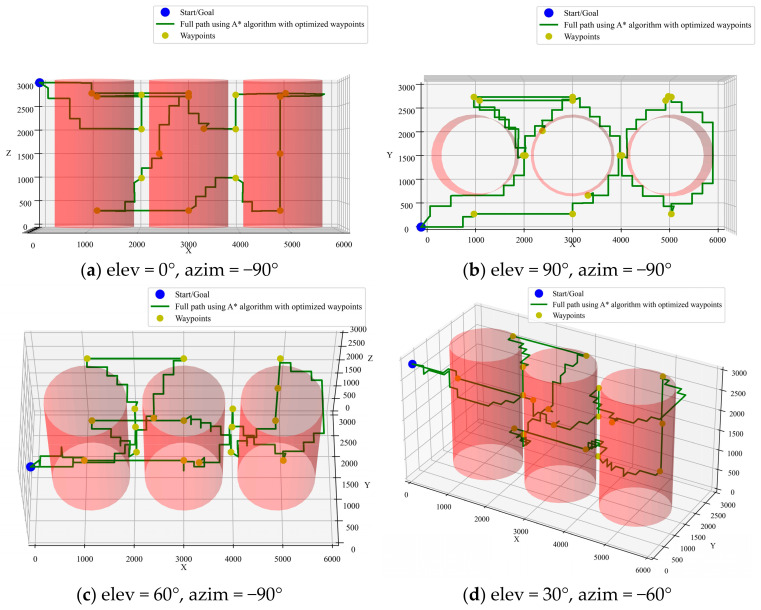
A* algorithm with ACO pre-planning.

**Figure 11 sensors-25-02666-f011:**
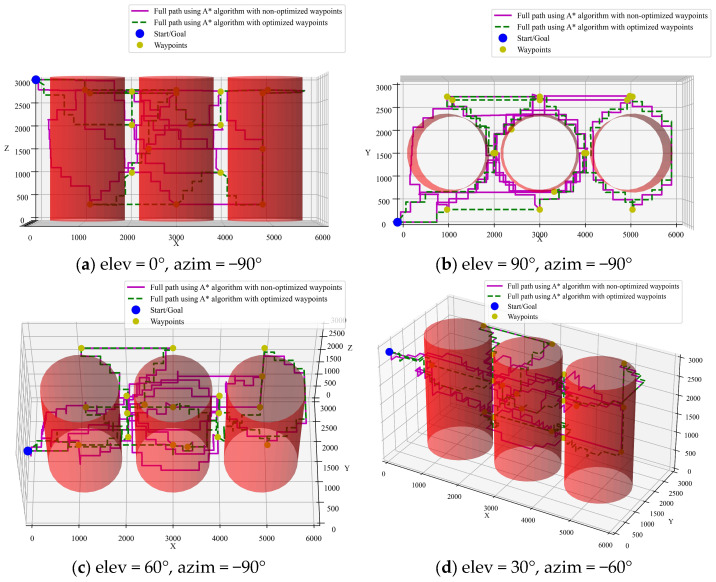
Comparison of A* algorithm optimized paths with and without pre-planning.

**Figure 12 sensors-25-02666-f012:**
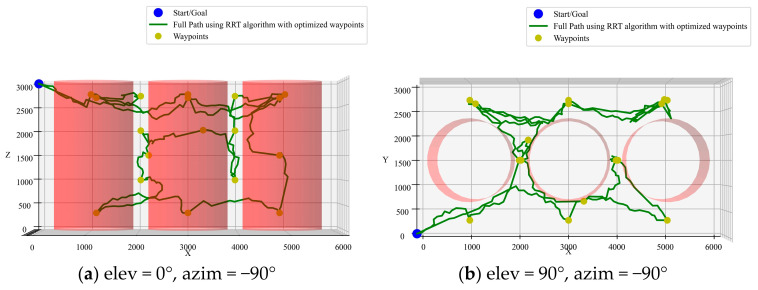
RRT algorithm with ACO pre-planning.

**Figure 13 sensors-25-02666-f013:**
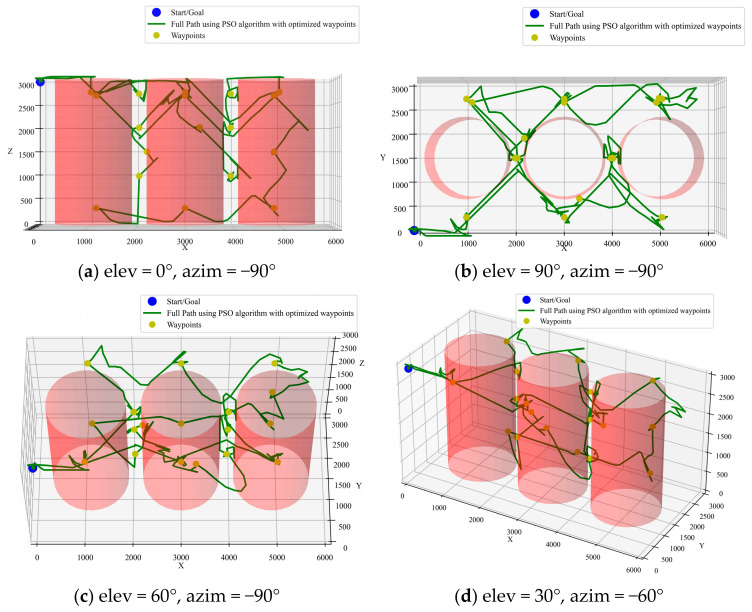
PSO algorithm with ACO pre-planning.

**Figure 14 sensors-25-02666-f014:**
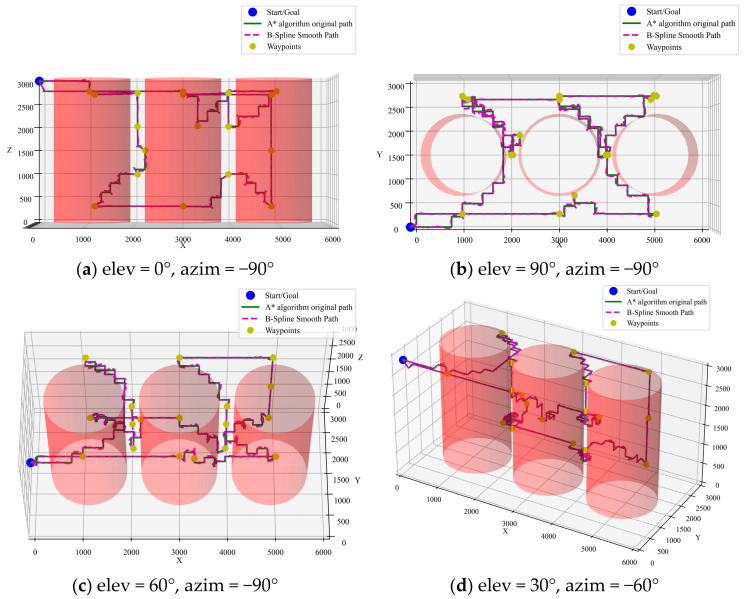
A* algorithm with ACO path smoothed by B-spline.

**Figure 15 sensors-25-02666-f015:**
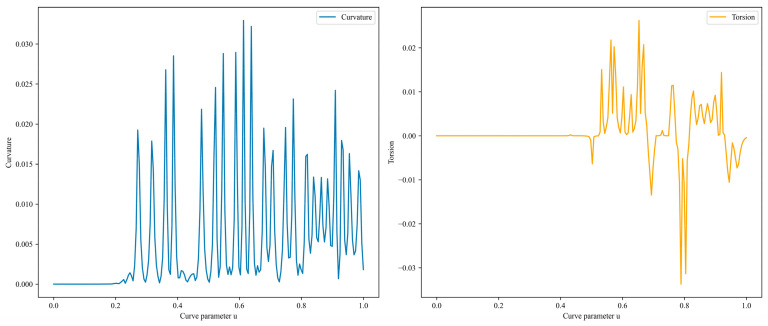
Curvature and torsion of the A* with ACO planned path after B-spline smoothing.

**Table 1 sensors-25-02666-t001:** Multi-point inspection locations and coordinates.

Inspection Points	Coordinates (mm)		
High-voltage lead-out detection points	(1100, 350,2700)	(3000, 350,2700)	(4900, 350,2700)
Low-voltage lead-out detection points	(1100, 2650, 2700)	(3000, 2650, 2700)	(4900, 2650, 2700)
Winding bottom and spacer detection point	(1100, 2650, 300)	(3000, 2650, 300)	(4900, 2650, 300)
Inter-winding detection points	(2050, 1500,1000)	(2050, 1500,2000)	(2050, 1500,2700)
	(3950, 1500,1000)	(3950, 1500,2000)	(3950, 1500,2700)
Historical fault points	(2200, 1900, 1500)	(3300, 700, 2000)	(4900, 2700, 1500)

**Table 2 sensors-25-02666-t002:** Path comparison of different algorithms.

Algorithms	Path Length (mm)	Features
A* with non-optimized waypoints	72,306	Longer path, good stability
A* + ACO	34,228	Shorter path, good stability
RRT + ACO	38,531~43,979	Shorter path, poor stability
PSO + ACO	64,163	Longer path, poor stability

**Table 3 sensors-25-02666-t003:** Numerical verification of control stability of A* + ACO hybrid algorithm path.

Parameters	A* + ACO Path	Smoothed Path	Improvement
Maximum curvature	0.12	0.033	79%
Max height error (*e_z_*, mm)	25.6	5.2	80%
Max lateral error (*e_xy_*, mm)	12.4	3.7	70.2%
Error convergence time	8.2s	2.1s	74.4%

## Data Availability

Dataset available on request from the authors.

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
