# Peer review of "Research on 3D Path Optimization for an Inspection Micro-Robot in Oil-Immersed Transformers Based on a Hybrid Algorithm"

_sensors, 2025, doi:10.3390/s25092666_

Round 1
Reviewer 1 Report
Comments and Suggestions for Authors
- The authors must indicate the time the robot can remain inside the transformer and whether that time is sufficient to perform a complete inspection. “While the micro-robot exhibits compactness and agility, its limited battery capacity necessitates the critical optimization of its 3D inspection path within the transformer.”
- Write: “The transformer contains three primary internal elements: electromagnetic coils, laminated cores, and insulation (dielectric mineral oil, paper, etc) that collectively enable efficient energy transfer through electro-magnetic induction.” Instead of: “The transformer contains three primary internal elements: electromagnetic coils, laminated cores, and dielectric mineral oil that collectively enable efficient energy transfer through electro-magnetic induction.”
- The authors do not indicate whether the robot can perform inspections in fluids other than mineral oil. “This necessitates the development of energy-efficient navigation algorithms capable of generating collision-free trajectories through the transformer’s labyrinthine interior.”
- The authors must indicate the characteristics of the transformers where it can be used (MVA, kV). “Therefore, this paper proposes an innovative hybrid algorithmic framework aimed at solving the path planning problem in a complex environment inside a large oil-immersed transformer.”
- What is the advantage of the robot having your geometry? “The body shell of the micro-robot is an elliptical sealed structure used to mount and protect the following functional modules.”
- Compare with other robots on the market, such as those from ABB. See the drive link indicated below. “The dimensions of the micro-robot are 15 × 15 × 26 cm (Length × Width × Height), which is determined for high throughput of the micro-robot in the narrow space of the transformer.”
- Explain how element 8 (Vertical propeller propulsion) in Fig. 1 operates.
- Indicate the units of Equation 3.
- What tests or improvements are needed for the robot to become commercial?
- Is it possible to design a robot that inspect the transformer while it is in operation?
- Include a list of symbols and acronyms at the end of the manuscript for ease of reading.
- Include the missing references in the manuscript from this file:

Reviewer 2 Report
Comments and Suggestions for Authors
Topic of fault inspection inside large transformers using micro-robots is very interesting, therefore, the work presented by the authors is very interesting, the optimization of the trajectories obtained with the different algorithms is adequate, however, I have the following comments:
Authors show the trajectories with the analyzed algorithms, with good results, but no detailed evidence is shown of the micro-robot inside the oil transformer, making the routes indicated in the different trajectories.
Authors show an image of the micro-robot, but do not provide technical details of its construction and operation, such as actuators, electronic circuits, control strategies, etc. Are the robot’s dimensions considered in the trajectories obtained?, How can we ensure that the robot can respond appropriately to the trajectories obtained? And how can we verify whether it is possible for the robot to perform the trajectories and turns in the narrow sections between the windings?
Authors mention the application of the B-spline neural network to smooth the trajectory, but no details of training and results are shown.
The authors report stability, however, they do not show how they calculate it and, above all, they do not show the corresponding analysis.
In the discussion section, the authors refer to the results as experimental results; however, they do not provide clear and detailed information on the experimental tests and the data acquisition process used to design the trajectories. The authors are encouraged to clarify whether the results are derived from modeling and simulation in 3D environments or whether the experimental tests were actually performed.
The authors mention trajectory optimization as a way to make the procedure more efficient and reduce energy consumption; however, they do not present an analysis of energy consumption, since the shortest trajectory does not necessarily result in the lowest energy consumption, as there may be greater joint intervention of the robot's actuators and therefore increases in energy consumption.
Round 2
Reviewer 2 Report
Comments and Suggestions for Authors
In letter to reviewers document, the authors have tried to explain in detail some terms to answer the observations made to them in the previous review.
In comment 4, the authors are mentioned as reporting stability; however, they do not show the analysis to determine that the system is stable; they only do so visually on the response in section 4.3. To report stability of the system, they would have to use one of the traditional methods, such as Routh's criterion, stability in the Lyapunov sense, etc.; the authors are recommended to use a more appropriate term.
In comment 6, the authors are asked whether the results are obtained through simulation or experimental testing. The authors have clarified that the results are obtained through simulations. They have even appropriately modified the term in the results section from experimental results to simulation results. In this regard, the authors are advised to clarify from the beginning of the document that the results they present are obtained through simulation.
